# Hydro-Geomorphologic-Based Water Budget at Event Time-Scale in A Mediterranean Headwater Catchment (Southern Italy)

**Albina Cuomo** [1,*] and **Domenico Guida** [1,2,*]

1    Department of Civil Engineering, University of Salerno, Via Giovanni Paolo II 132, 84084 Fisciano, Italy
2    interUniversity Research Center for the Prediction and Prevention of Major Hazards (C.U.G.RI.), University of Salerno, 84084 Fisciano, Italy
*    Correspondence: acuomo@unisa.it (A.C.); dguida@unisa.it (D.G.)

**Abstract:** The Ciciriello catchment is a 3 km$^2$ drainage sub-basin of the Bussento river basin, located in the southern part of the Campania Region (Southern Italy). Since 2012, this catchment has been studied using an interdisciplinary approach—geomorphological, hydrogeological, and hydrological— and a hydro-chemical monitoring system. Following previous research, the aim of this paper is to calibrate, on this catchment, the hydrologic parameters for a water budget at event time-scales using the HEC-HMS model, adopting object-based hydro-geomorphological class features. Firstly, lumped modeling was performed to calibrate the hydrologic parameters from 20 observed hydrographs at the downstream monitoring station of the Ciciriello catchment. Then, physical-based rainfall–runoff modeling was conducted using three different procedures: (1) applying the recession coefficients to each outlet with a newly defined hydro-geomorphologic index (HGmI); (2) assessing the storage coefficient for each sub-basin as a weighted mean of HGmI; and (3) using the storage coefficient associated with the largest HGmI in the sub-basin. The adopted procedures were tested using diverse goodness-of-fit indices, resulting in good performance when the object-based hydro-geomorphotypes were used for the parameter calibration. The adopted procedure can thus contribute to improvements in rainfall–runoff and water budget modeling in similar ungauged catchments in Mediterranean, hilly, and forested landscapes.

**Keywords:** water budget; HEC-HMS; hydro-geomophology; headwater; hydro-geomorphologic index

## 1. Introduction

A widely discussed issue in hydrology and fluvial geomorphology concerns the rainfall–runoff transformation modeling of ungauged catchments that need to be set up in the field for flood hazard assessment [1,2], catchment management applications, or simply for understanding the catchment's functioning [3–5] and how individual processes combine to produce the overall catchment response [6–8]. To achieve these aims, different rainfall–runoff models are commonly used, but they need many input parameters, from physical-based to calibration parameters [9]. The former can be observed or estimated from easily detectable parameters, while the latter are generally back-calculated from rainfall– runoff data analysis [7]. The prediction of ungauged catchments that are not monitored requires improving the relationships between model parameters and easily obtainable information, such as topography, geology, landforms, vegetation, and soils [10].

The advent of improved spatial data sources and tools to handle information about land use, topography, vegetation, ecology, and morphology has enabled a number of authors to suggest various combinations of land-surface characteristics that can be used to define areas of similar hydrological responses. Some authors have described a catchment disaggregation approach that involves the subdivision of regional-scale catchments into a

number of hydro-types with similar land-use characteristics, slopes, and elevations [11,12]. A catchment disaggregation approach based on distinct vegetative characteristics was also described in the work of Liang et al. [13], while Jain et al. [14] divided the catchment into a number of hydro-types according to elevation and land cover information. The work of Beker and Jaun [15] assessed up to nine different areal disaggregation schemes based on land-use, land cover (vegetation), soil type, and slope class for a small-scale river basin.

From the previous research, it is evident that the hydro-type disaggregation method can overcome the critical effects of averaging associated with lumped land-surface representations, in addition to being more realistic in terms of data requirements and computational time compared to the distributed modeling approach [16]. In hydrology, Wood et al. [17] developed the representative elementary area (REA) in the order of 1 km$^2$ using a hypothetical study of the effects of variable topography, soils, and rainfall, but only for short rainfall correlation lengths. In 1998, conservation equations for mass, momentum, energy, and entropy were formulated for a watershed that was divided into smaller discrete units called representative elementary watershed (REW) [18]. Recently, the REW method was updated with the mass balance equation associated with the subsurface storm flow, and the new model code named REWASH, was revised accordingly [19].

The quasi-distributed variable infiltration capacity (VIC) hydrological model was developed in an attempt to reproduce a larger-scale hydrological response. The VIC model incorporates the saturation–overland flow mechanism with a continuous probability density function (PDF) to describe the relationship between soil moisture content and saturation, with relevant hydrological quantities determined via integration over this distribution. The sensitivity of the simulated runoff to the parameters that control its generation in the VIC model was investigated by Demaria et al. [20]. More recently, a catchment-specific sensitive parameter calibration for streamflow simulations in the VIC model was performed [21]. Despite the need for calibrating the VIC parameters, this model is largely employed for physical-based modeling [22–25] in different regions.

In 2000, Sidle [3] worked on developing a hydro-geomorphic paradigm to describe stormflow generation in steep headwater catchments, introducing the concept of geomorphic units and including zero-order basins, channels, riparian zones, and hillslopes. Each hydro-geomorphic unit that has unique characteristics and flow paths associated with stormflow generation was modeled by Kim et al. [5] by modifying the simple tank model into a multi-tank model for zero-order basins and hillslope simulations. In addition, to capture flow dynamics in the riparian corridor and route water in the catchment, Sidle et al. [4] implemented a kinematic wave model for the multi-tank model.

Following the hydro-geomorphic paradigm, Cuomo [26] introduced and applied a new hydro-geomorphological basic unit, the hydro-geomorphotype (HGmT), using the Salerno Geomorphological Mapping System [27,28] as a framework for object-based geomorphological mapping. Then, the identification and delimitation of hydro-geomorphotypes, as the basic unit in interdisciplinary studies on water resource and flood hazard assessment, planning, and management, were proposed by Cuomo [26]. The proposed approach, developed via GIS-based procedures (firstly using grid-based and successively using object-based methods (e-Cognition software) with the time–space, multi-scale, and hierarchy principles), was calibrated and validated in experimental basins located in the Cilento Geopark (Campania region of Italy).

A recent study focused on the use of the HGmT to drive a specific hydro-geomorphological and hydro-chemical monitoring program that was applied to small, forested headwater catchments to detect the sources, pathways, and timing of the different runoff components [29]. Based on the results of Cuomo and Guida [29], where a novel mass balance was introduced to separate the hydrograph into four runoff components, Guida et al. [30] highlighted a new procedure for identifying and separating hydro-chemical runoff components and a geomorphometric application for the objective delimitation of the source areas, from which each runoff component is generated [31,32].

In this paper we propose a procedure to calibrate the hydrologic parameters for a water budget at the event time-scale in a study area located in the southern part of the Campania Region (Southern Italy)—the Ciciriello catchment. This research, performed using the HEC-HMS model and the HGmT (individuated and delimitated at regional scale by Cuomo [26] in the Campania region) includes three steps: calibrating the hydrologic parameters, physical-based rainfall–runoff modeling, and testing the adopted procedures.

In this paper we first describe the study area and the dataset used for the research, then we focus on the hydrologic method and a description of the hydro-geomorphologic approach. To conclude, the results of the physical-based rainfall–runoff generation and water budget models at the outlet of the Ciciriello are presented along with related discussions.

## 2. Materials and Methods

### 2.1. Study Area

The Ciciriello research catchment (LAT. 40.1957 LONG. 15.5379) is located in the southeast of Campania region (Southern Italy), within the territory of the UNESCO Cilento Global Geopark. The altitude of the catchment ranges from 420 to 812 m above sea level, and its area is about 3 km$^2$. The catchment is characterized by a Mediterranean climate, typical of the Southern Tyrrhenian Borderland. The average annual rainfall is around 1400 mm, with a marked difference between the summer monthly rate of around 30 mm and the average winter monthly rate of around 250 mm [33].

The experimental catchment is an exclusively terrigenous bedrock outcropping watershed. At the base, a marly–clayey formation passes by an evident incoformity upward to a southwestern dipping sandstone sequence [34]. Along the left valley side, inter-bedded into the sandstone strata, one can find up to 10 m thick lenticular marly bed outcrops (Figure 1). On the upper ridges, the bedrock is covered by thick Regosols, regolith on the noses and spurs, and gravelly slope deposits at the toes of the open slopes [35]. Along the left valley side, the underlying rock strata are dip sloping (cataclinal valley slope) in areas where shallow flow-like landslides occur (Figure 1), whereas an anti-dip slope (anaclinal valley slope) is detected along the right valley side and is influenced by deep-seated mass movements.

The mainstream bed, consequent to dipping strata in addition to the main faults, is incised partly in alluvial, coarse deposits and partly on bedrock.

In the headwaters, colluvial hollows are located in the bottom of the zero-order basins. Permanent springs from bedrock aquifers and seasonal outflows from colluvial swallets increase progressively downstream of the stream discharge.

To define the runoff source areas, an object-based hydro-geomorphological map of the Ciciriello catchment was created by Cuomo and Guida [29] (Figure 2). Starting from the Campania Region Technical Map (CTR), at 1:5000, a digital elevation model (DEM) with a 5 m cell size was obtained and used for the object-based hydro-geomorphological mapping, delimiting the HGmT.

From a hydro-geomorphological point-of-view, the attitude of the bedrock formation boundary controls the overall groundwater circulation, and the lower-stand marly–clayey formation constitutes the local aquitard below the upper-stand sandstone aquifer. The westward dipping of the permeability boundary induces a conformed general westward groundwater flow, locally convergent toward the lower apex of the "hydro-wedge", as introduced in Cascini et al. [36], where the main permanent springs are located. In the headwaters, colluvial hollows cover the bottom of the zero-order basins (ZOBs), which can be considered the main headwater HGmT [26], where dominant excess saturation runoff occurs, mainly during the wet season, from October to March.

### 2.2. Dataset

Since 2012, the research catchment has been equipped for monitoring water depth (D), discharge (Q), and electrical conductivity (EC). A Swoffer 3000 current meter (Swoffer Inc., Sumner, WA, USA) has been used for discharge measurements, whereas a multi-

parametric probe (HI 9828 Hanna Instruments Inc., Nusfalau, Romania) has been used for EC measurements. During selected storm events, 10 min D and EC data are recorded at the main station using a data logger DL/N70-Multi (STS Inc., Sirnach, Switzerland). Rainfall data at a 10 min time resolution for the Sanza rain gauge were provided by the Civil Protection Service of the Campania Region. A number of monitoring stations were located along the drainage network of the Ciciriello catchment to better understand the hydrologic and hydrologeologic behavior of the catchment [29]. The main monitoring station is located at the outlet of the catchment (420 m above sea level), where the parameter data have been collected every day since before 2015. The location and timing of the monitoring activity is based on detailed, multi-temporal hydro-geomorphological surveys and mapping, oriented by the variable source areas concept [37] and the above citied hydro-geomorphic paradigm [3].

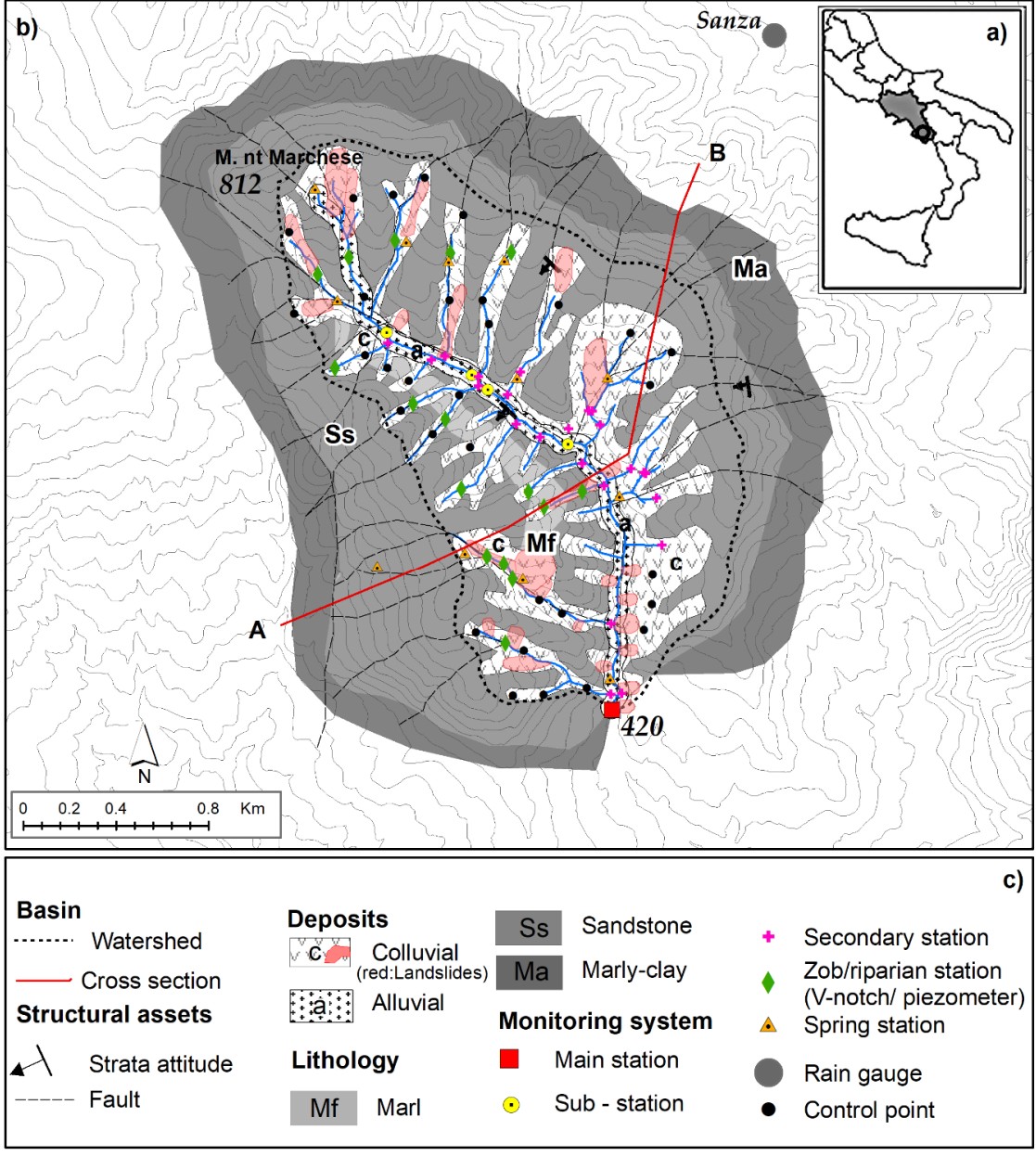

**Figure 1.** Ciciriello experimental catchment located in the southeast of Campania (Southern Italy) (modified from [35]). (**a**) Upper right inset: geographical location of the Bussento river basin; (**b**) left inset: schematic geological map, hydro-structural features, and location of the monitoring stations; (**c**) legend.

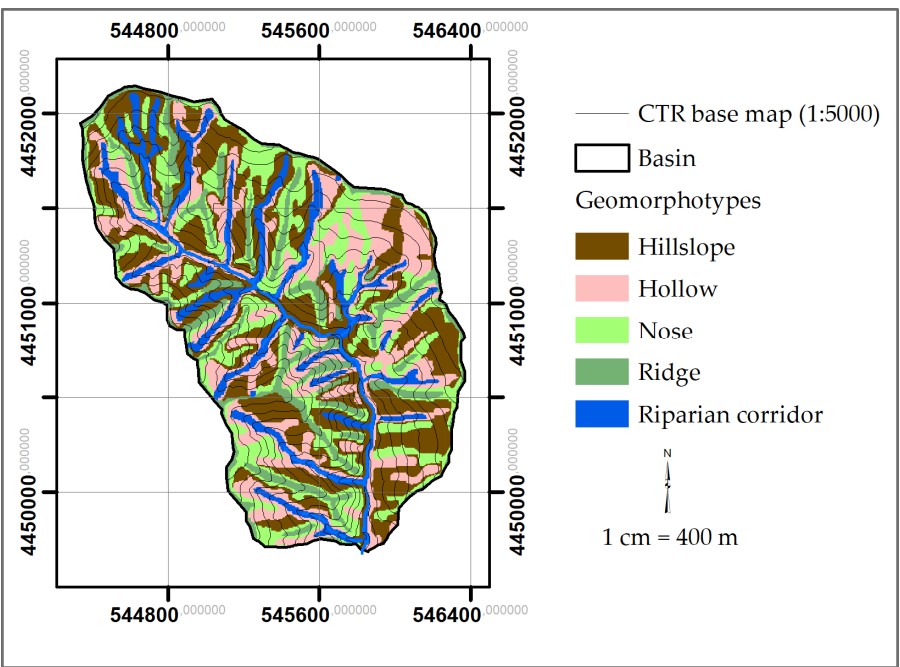

**Figure 2.** Object-based geomorphological map of the catchment depicting the geomorphotypes. (modified from [30]).

Figure 3 depicts the daily discharge measured at the main station and the precipitation of the hydrologic year (starting from October) of 2012–2013 recorded by the Sanza rain gauge. The black box highlights the rain events explored for the rainfall–runoff simulation performed in our study.

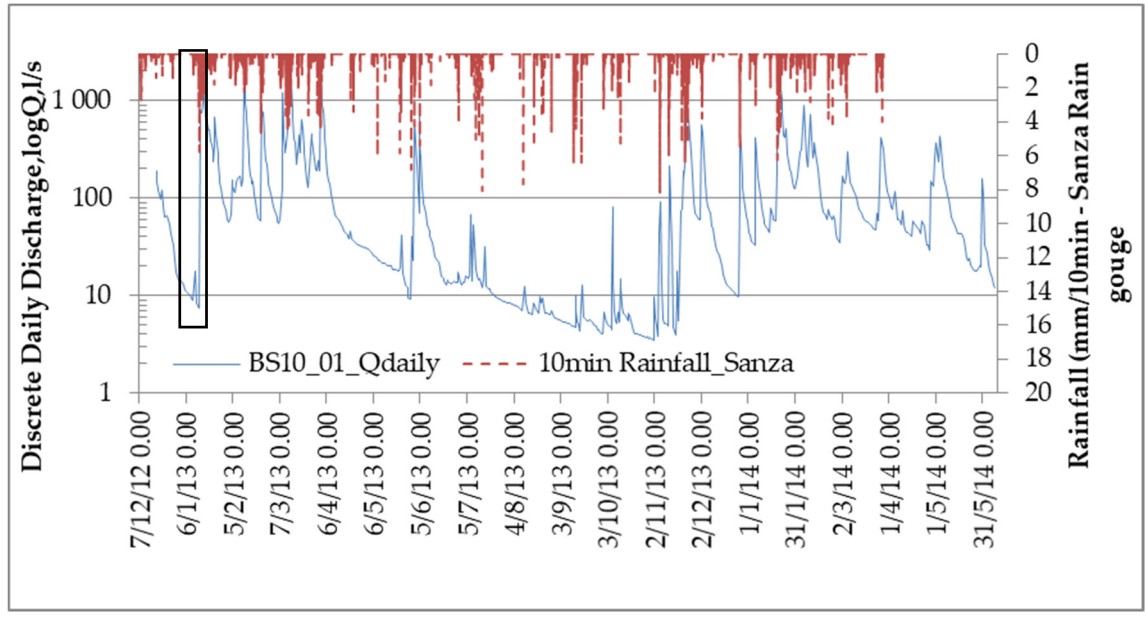

**Figure 3.** Discrete daily discharge collected at the main station of the Ciciriello catchment and the rain plot from the Sanza rain gauge (10 min time intervals).

The event used for the simulation occurred on 1 April 2013. This event was extracted from multiple events at the end of the wet period when the soil was saturated, and about 90% of the volume flow was transformed into runoff volume. The antecedent precipitation

index (API) values calculated at 15 and 30 days before the peak (Table 1) were 53 and 108 mm, respectively.

**Table 1.** Storm flow parameter descriptions: $IP_{max}$, maximum precipitation intensity; Pt, total precipitation; $Q_o$, discharge before the beginning of the rising limb; $Q_{max}$, peak flow; API: antecedent precipitation index 15 and 30 days before the peak; RC, runoff coefficient.

| Storm Event | $IP_{max}$( mm/h) | Pt (mm) | $Q_o$ (l/s) | $Q_{max}$ (l/s) | API15 (mm) | RC * |
|---|---|---|---|---|---|---|
| 1/4/13 10.00 | 4.8 | 65.8 | 189 | 2200 | 53 | 0.9 |

* the value was calculated with respect to the areas that actually contributed to the runoff generation (the riparian corridor, the hillslopes, and the headwater. The summit areas were not considered).

*2.3. Hydrologic Modeling*

Hydrologic modeling concerns the discharge hydrograph simulation at the outlet of the Ciciriello catchment, where observed daily discharge and precipitation data are available. For the simulation, we adopted two procedures. In the first, for which a lumped simulation was performed, the parameters were estimated from events selected from the 2012–2013 hydrologic year and then employed for the event discharge simulation. In the second, a distributed model was adopted, by dividing the Ciciriello catchment into 23 ungauged sub-basins. Therefore, the lack of local runoff data that could be used for calibrating the model parameters was overcome by using the relationship between the hydrologic parameters and the geomorphologic parameters.

For the event-based hydrologic simulation, the HEC-HMS model was used. This model is a physical-based, semi-distributed hydrologic model designed by the U.S. Army Corps of Engineers to simulate the hydrologic cycle processes for a variety of catchments [38]. This model can be used for simulating a single storm event, which may range from a few hours to a few days, or a long-term period of stream flow (daily, monthly, and seasonal) [38].

HEC-HMS can be applied as a lumped or distributed rainfall–runoff model, suitable for small and large catchment hydrologic applications [39], water balance studies [40], analysis of the impact of land use and climate change on runoff generation [41], and flooding [42]. It is suitable for simulations in ungauged basins because many of the modules used in the HEC-HMS contain parameters with a physical basis that can be estimated from measurable properties of the watershed.

To run the HEC-HMS hydrologic simulation, one needs to specify three datasets:

1. The Basin Model: This contains the physiographic representation of the watershed and can be managed under the Arc Hydro tool in the ArcMap software. For distributed modeling, the drainage network was divided into segments, and the catchment was split at the end of each segment into 23 sub-catchments.

2. The Meteorologic Model: It includes meteorological data of the input for the rainfall and evapo-transpiration. The simulated event was derived here from a rain input that occurred on 1 April 2013, recorded at the station of Sanza village with 10 min time intervals.

3. The Control Specification storage for all the input datasets: Where, the temporal range used for the calculation and the output to simulate the hydrograph was the same for the rainfall.

The simulation of the runoff produced at the main station from the rain event of 1 April 2013 was performed using the following equations:

- The Lag method for simulating the channel flow, such that the outflow hydrograph was similar to the inflow but lagged in time. The lag time for a sub-catchment was assumed to equal 0.6 times the time of concentration.

- The recession method for the baseflow simulation. The baseflow represents the sustained runoff of previous precipitation stored temporarily in the catchment into the channel. In this study, we adopted the recession method for simulating the baseflow. The necessary routing parameters were evaluated from about 20 observed runoff hydrographs recorded at the main station during the 2012–2013 year. On the recession curve (that is,

the lower part of the falling limb of a hydrograph), the Maillet equation with a simple exponential relationship was used [43]:

$$Q_t = Q_0 e^{-\alpha t} = Q_0 k t \tag{1}$$

where $Q_t$ is the discharge at time t, $Q_0$ is the initial discharge, $k = e^{-\alpha t}$ is the recession constant, and $\alpha$ is the recession rate. For the $\alpha$ calculation, the recession limb of the discharge hydrograph was plotted on a logarithmic scale, so the recession curves can generally be classified into several segments based on the inflection points, which are indicative of a transition in the drainage structure. In this way, the major sources and storage systems, such as overland and subsurface flows, can be inferred [44]. The required parameter for the simulation is $\alpha$, which, for the lumped model, was assumed to be the mean value of the 20 events analyzed. For the distributed model, the $\alpha$ was instead calculated at the outlet of the sub-basins using the HGmI, as described in the following.

Clark's method was used to transform the rainfall into runoff. The HEC-HMS computes the runoff volume through the synthetic unit hydrograph (UH) specification. Among the available UHs in the model, we used Clark's UH, which is a quasi-conceptual UH that accounts for watershed storage. The Clark unit hydrograph method explicitly represents two critical processes for the translation of excess rainfall ($t_c$) and attenuation due to the effects of storage in the sub-basins (R). As noted, the linear routing model properties are defined implicitly by a time–area histogram; the typical time–area relationship used in the HEC-HMS model is

$$\frac{A_t}{A} = \left\{ \begin{array}{ll} 1.414 \left(\frac{t}{t_c}\right)^{1.5} & \text{for } t \leq \frac{t_c}{2} \\ 1 - 1.414 \left(1 - \frac{t}{t_c}\right)^{1.5} & \text{for } t \geq \frac{t_c}{2} \end{array} \right\} \tag{2}$$

where At = the cumulative watershed area contributing at time t; A = the total watershed area; and $t_c$ = the time of concentration of the watershed. Application of this implementation only requires the parameter $t_c$, the time of concentration, which can be estimated via calibration. In the present work, the $t_c$ parameter was calculated by applying the Giandotti formula:

$$t_c = \frac{4\sqrt{A_b} + 1.5 L_a}{0.8\sqrt{z_m - z_0}} \tag{3}$$

where $t_c$ = the time of concentration; $A_b$ = the total watershed area (km$^2$); $L_a$ = the length of the main channel (km); $z_m$ = the mean height of the watershed (m); and $z_o$ = the height at the outlet (m). For the distributed simulation, the tc was calculated for each sub-basin by applying Equation (3), which contains readily available physiographic parameters.

The basin storage coefficient, R, is an index (with units of time) of the storage time of excess precipitation in the watershed as it drains to the outlet point. This index can also be estimated via calibration if gauged precipitation and streamflow data are available, or the R can be computed. For this research, the storage coefficient was estimated from the observed discharge hydrograph recorded at the catchment outlet by applying Equation (1). The R was computed for each runoff component, i.e., direct, fast, delayed, and shallow, by plotting the recession curve in a semi-logarithmic plot and applying Equation (1) to each highlighted segment. Each segment identifies a specific runoff process, one of which is associated with the corresponding hydro-geomorphic unit (e.g., nose, hollows, riparian areas, and hillslopes).

Moreover, the losses in evapotranspiration can be neglected due to the event scale simulation. The losses of infiltrations are also neglected because the simulated event occurred at the end of the wet period, when the soil was saturated.

### 2.4. Hydro-Geomorphologic Modeling

The distributed model application requires simulation of the discharge hydrograph at the outlet of each sub-basin in which the Ciciriello was divided. To this end, the conceptual

models used in the HEC-HMS for the simulation rely on empirical data to make predictions about water movement [45]. Nevertheless, many of these models contain parameters with a physical basis and can be estimated from measurable properties of the watershed. Measurable properties of the watershed are the geomorphologic parameters that can be retrieved from topographic data, satellite images, or direct geomorphological surveys. In addition, these parameters are directly linked to the hydrologic behavior of the catchment. Indeed, the geomorphology influences the hydrology, and vice versa [35]. In this paper, the HGmT was used for optimizing the hydrologic simulation where the R and $\alpha$ parameters were not available at the outlets of the 23 sub-basins.

The HGmT map production started from an object-based geomorphological map of the catchment (Figure 2) and for each, the "combine" tool of ArcMap was used to infer the corresponding hydrologic, starting from specific EC end-member values for each HGmT, as shown in Table 2.

**Table 2.** Landform vs. hydro-geomorphotype correspondences (modified from [35]).

| Landform, Component, or Element | Geomorphotype [27] | Hydro-Geomorphological Behaviour | Hydro-Geomorphotype (HGmT in [26]) | EC Range (μS/cm) [35] |
|---|---|---|---|---|
| Upland, summit, peak, crest | Ridge | Groundwater recharge on bare bedrock and dominant excess infiltration runoff after storm | Deep percolation | 250–300 <100 |
| Shoulder, side slope | Nose | Shallow soil, groundwater recharge area, prevalent excess infiltration runoff | Deep percolation | 250–300 <100 |
| Scarps, back-slope, foot-slope, wash-slope, talus | Hillslope | Debris, deep soil, shallow aquifer, excess saturation excess and sub-surficial runoff | Fast return flow | 120–180 |
| Glen, swallet, scar | Hollow | Deep soil, shallow aquifer, prevalently excess saturation, delayed runoff production | Delayed return flow | 200–220 |
| V-shaped stream, gully, bank, stream bed | Riparian corridor | Shallow soil, groundwater discharge, prevalently sub-surface, delayed return flow and groundwater ridging | Direct | 80–120 |

For the tc and R parameter assignment at the sub-basin outlet, the empirical relationship with the HGmT was used.

To this end, the R value, calculated at the main station from observed hydrographs for the four components (direct, fast return flow, delayed return flow, and deep percolation), was estimated at the outlet of each sub-basin by adopting the following equation:

$$R_{sub-basin} = \sum \frac{R \times A_{sub-basin(HGmT)}}{A_{sub-basin}} = \sum R \times HGmI \qquad (4)$$

where $R_{sub-basin}$ = the storage coefficient calculated at the outlet of the sub-basin for a specific component (direct, sub-surface flow, or deep percolation); $A_{sub-basin}$ = the total area sub-basin (m²); R = the storage coefficient calculated at the main station for a specific component; and $A_{sub-basin\ (HGmT)}$ = the total area of hydro-geomorphotype for the considered runoff component in the sub-basin (m²). The ratio between the two areas was named the "Hydro-Geomorphotype index" (HGmI) as a significant parameter providing useful information about the percentage of the catchment area affected by a specific runoff process connected to the HGmT's contribution. To obtain the best simulation, R was also calculated while considering only the main hydro-geomorphotype. At the outlet of the sub-basin, R was assigned from the max HGmI calculated in each sub-basin.

For the recession coefficient calculation at the outlet of each sub-basin, the following formula was adopted:

$$\alpha_{\text{sub-basin}} = \frac{\alpha \times A_{\text{sub-basin(DP)}}}{A_{\text{sub-basin}}} = \alpha \times \text{HGmI}_{\text{dp}} \tag{5}$$

where $\alpha_{\text{sub-basin}}$ = the recession coefficient at the outlet of the sub-basin (1/t); $\alpha$ = the recession coefficient estimated at the main station; $A_{\text{sub-basin(DP)}}$ = the area delimited as deep percolation in the sub-basin ($m^2$); and $A_{\text{sub-basin}}$ = the area of the sub-basin ($m^2$). The $\alpha$ parameter was calculated with respect to the HGmI derived from the areas classified as deep percolation (dp).

*2.5. Model Perfomance Evaluation*

The performance of prediction models can be assessed using a variety of different methods and metrics. Here, to evaluate the model performance and to determine if the hydro-geomorphologic procedure can effectively give insights into the rainfall–runoff modeling of ungauged catchments, the following indices were calculated:

1.  Mean Absolute Error (MAE):

$$\text{MAE} = \frac{1}{n} \sum_1^n \left| Q_{\text{mod,i}} - Q_{\text{obs,i}} \right| \tag{6}$$

2.  Mean Squared Error (MSE)

$$\text{MSE} = \frac{1}{n} \sum_1^n \left( Q_{\text{mod,i}} - Q_{\text{obs,i}} \right)^2 \tag{7}$$

3.  Root Mean Squared Error (RMSE)

$$\text{RMSE} = \left[ \frac{1}{n} \sum_1^n \left( Q_{\text{mod,i}} - Q_{\text{obs,i}} \right)^2 \right]^{1/2} \tag{8}$$

4.  Nash–Sutcliffe Efficiency coefficient (NSE)

$$\text{NSE} = 1 - \frac{\sum_1^n \left( Q_{\text{obs,i}} - Q_{\text{mod,i}} \right)^2}{\sum_1^n \left( Q_{\text{obs,i}} - \overline{Q}_{\text{obs,i}} \right)^2} \tag{9}$$

5.  Index of agreement (d)

$$d = 1 - \frac{\sum_1^n \left| Q_{\text{mod,i}} - Q_{\text{obs,i}} \right|}{\sum_1^n \left( \left| Q_{\text{mod,i}} - \overline{Q}_{\text{obs,i}} \right| + \left| Q_{\text{obs,i}} - \overline{Q}_{\text{obs,i}} \right| \right)} \tag{10}$$

where $Q_{\text{obs}}$ = the observed discharge; $Q_{\text{mod}}$ = the modeled discharge, and $Q_{\text{obs}}$ = the mean observed discharge.

## 3. Results and Discussions

*3.1. Recession Curve Analysis*

The hydrologic parameters useful for modeling were computed at the main station, where discharge hydrographs have been available since 2012. A streamflow hydrograph recession analysis was then performed on selected 20 events. By plotting the recession curve in a semi-logarithmic plot, three segments were clearly identified, each representative of a specific runoff process. Only for a few events was a fourth segment was identified. This condition derived from the monitoring time-step, since the daily discharge was not always able to detect the direct runoff. Therefore, considering four segments, the recession coefficients were named from largest to smallest as $\alpha_1$, $\alpha_2$, $\alpha_3$, and $\alpha_4$ (Table 3).

**Table 3.** Recession coefficient (1/day) descriptive statistics.

| $\alpha$ (1/Day) | Mean | Max | Min | SD |
|---|---|---|---|---|
| Direct runoff ($\alpha_1$) | 5.40 | 6.90 | 2.50 | 2.05 |
| Quick return flow 1 ($\alpha_2$) | 0.52 | 0.80 | 0.36 | 0.14 |
| Delayed return flow ($\alpha_3$) | 0.161 | 0.21 | 0.11 | 0.038 |
| Deep percolation ($\alpha_4$) | 0.04 | 0.09 | 0.02 | 0.02 |

The largest recession coefficient ($\alpha_1$) had a mean value of 5.40 1/day (about 5 h). The medium values, $\alpha_2$ and $\alpha_3$, were, respectively, 0.52 (2 day) and 0.16 1/day (6 day), whereas the smallest recession rates ($\alpha_4$) had a mean value of 0.04 1/day (about 30 day). The obtained $\alpha$ coefficients are comparable with previous results in the same catchment [31,35,46].

The interpretation of these values in terms of runoff generation was derived from [29], where hydro-chemical analysis confirmed four runoff generation mechanisms, and from Guida et al. [35], where a perceptual model of Ciciriello was applied. According to these studies, the initial recession coefficient ($\alpha_1$), as a result of its fast response to the rain input, showed a steep segment in the upper part of the recession curve. This shape also reflects the rapid exhaustion of the direct runoff. The other parts of the curve had lower recession coefficients ($\alpha_2$ and $\alpha_3$) because the great water volumes stored in the top and sub-soils were released by means of soil pipe flows ($\alpha_2$) and subsurface-tile drainage systems ($\alpha_2$).

The last segment shows the lowest slope and has the smallest recession rate ($\alpha_4$); here, the water is stored in shallow aquifers and hydro-wedges and is slowly draining into the river system.

The time needed for water to be stored in the soil before being able to reach the stream river was derived based on the recession constant in Table 3. The storage times for the direct runoff, the fast and delayed return flow, and the deep percolation are listed in Table 4.

**Table 4.** Storage time for the main runoff mechanisms detected in the Ciciriello catchment.

| Runoff Mechanisms | t (h) |
|---|---|
| Direct runoff | 5.26 |
| Quick return flow | 49.61 |
| Delayed return flow | 157.51 |
| Deep percolation | 729.76 |

The storage time of the deep percolation was used as an input parameter for the recession modeling in HEC-HMS at the main station and for deriving the recession constant at the outlet of the sub-basins by applying Equation (5). The other t values were used for calculating the input R in Clark's UH module for both the main station and sub-basins.

### 3.2. Lumped-Based Hydrologic Simulation

Lumped-based modeling was performed to obtain the initial calibration of the hydrological parameters to be used in the subsequent distributed modeling. To this end, the simulations use diverse parameter values to optimize the final modeled hydrograph. The $t_c$ value, calculated from empirical Equation (3) and related to the catchment's $t_c$ value, was assumed to be constant during the simulations. Table 5 outlines all the necessary parameters for the estimation of tc and the resulting $t_c$ after applying Equation (3).

**Table 5.** Morphometric parameters of the Ciciriello catchment. Ab: watershed area; La: length of the main channel (km); $z_m$ = mean height of the watershed (m); $z_o$ = height at the outlet (m); and $t_c$ = the time of concentration of the watershed.

| Ab (km$^2$) | La (km) | Zm (m) | Z0 (m) | tc (h) |
|---|---|---|---|---|
| 3.04 | 2.53 | 617.15 | 393.6 | 0.9 |

Using the $\alpha$ intervals of Tables 3 and 6, several simulations were run while changing the R value to 5, 13, 15, 20, 50, and 150 h (Figure 4). The baseflow $\alpha$ value was assumed to be 0.04 1/day (see Table 3) and was used for the specific recession module in the HEC-HMS simulation.

**Table 6.** Morphometric parameters of Ciciriello's sub-basins. $A_b$: watershed area; $L_a$: length of the main channel (km); $Z_m$ = mean height of the watershed (m); $Z_o$ = height at the outlet (m); and $t_c$ = the time of concentration of the watershed.

| Sub-Basin | Channel | $A_b$ (km²) | $L_a$ (km) | $Z_m$ (m) | $Z_0$ (m) | $t_c$ (h) |
|---|---|---|---|---|---|---|
| W240 | R20 | 0.167 | 0.449 | 715.89 | 583.5 | 0.25 |
| W250 | R40 | 0.155 | 0.356 | 705.48 | 579.2 | 0.23 |
| W260 | R10 | 0.153 | 0.205 | 682.23 | 582.9 | 0.23 |
| W270 | R30 | 0.002 | 0.019 | 597.32 | 579.4 | 0.06 |
| W280 | R60 | 0.134 | 0.210 | 664.31 | 541.4 | 0.20 |
| W290 | R50 | 0.186 | 0.415 | 619.54 | 541.2 | 0.33 |
| W300 | R80 | 0.132 | 0.250 | 667.16 | 528.3 | 0.19 |
| W310 | R120 | 0.190 | 0.378 | 647.95 | 520.1 | 0.26 |
| W320 | R70 | 0.021 | 0.166 | 562.50 | 527.8 | 0.18 |
| W330 | R150 | 0.306 | 0.415 | 641.24 | 499.0 | 0.30 |
| W340 | R90 | 0.001 | 0.028 | 532.60 | 526.8 | 0.07 |
| W350 | R100 | 0.110 | 0.088 | 637.45 | 526.8 | 0.17 |
| W360 | R110 | 0.018 | 0.130 | 549.44 | 519.3 | 0,17 |
| W370 | R130 | 0.028 | 0.122 | 565.89 | 514.0 | 0.15 |
| W380 | R160 | 0.156 | 0.222 | 635.06 | 514.2 | 0.22 |
| W390 | R140 | 0.087 | 0.341 | 558.01 | 497.9 | 0.27 |
| W400 | R170 | 0.178 | 0.312 | 587.31 | 474.0 | 0.25 |
| W410 | R180 | 0.171 | 0.150 | 607.88 | 474.3 | 0.20 |
| W420 | R200 | 0.341 | 0.586 | 536.61 | 426.8 | 0.38 |
| W430 | R190 | 0.228 | 0.514 | 602.15 | 427.7 | 0.25 |
| W440 | R220 | 0.093 | 0.361 | 460.84 | 396.3 | 0.27 |
| W450 | R210 | 0.179 | 0.410 | 572.33 | 396.7 | 0.22 |
| W460 | R230 | 0.006 | 0.066 | 408.43 | 393.6 | 0.13 |

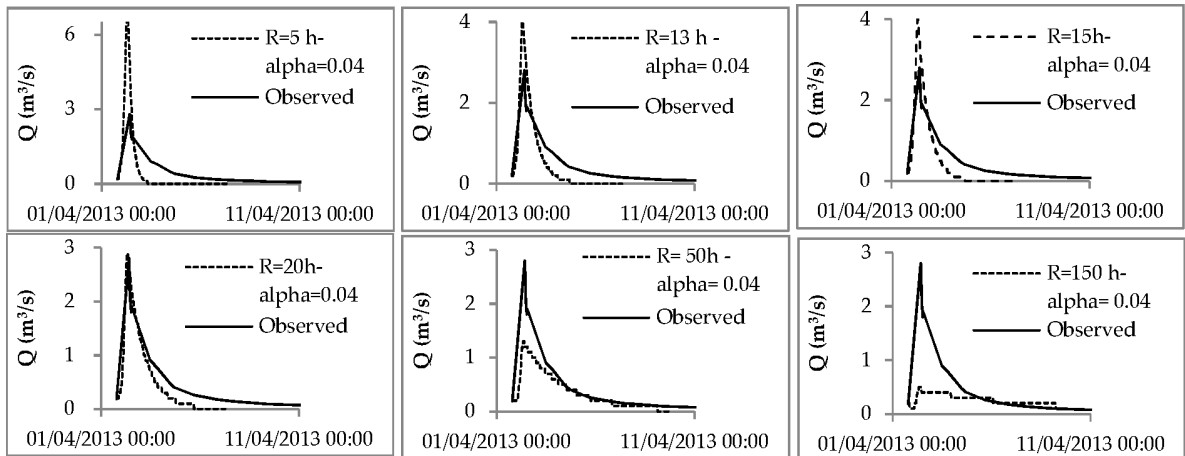

**Figure 4.** Lumped-based rainfall–runoff modeling performed by changing the R value to 13, 15, 20, 50, and 150 h.

The simulations highlighted the model's sensitivity to R value variation. The R values used for the runoff simulation fell within the value ranges calculated for the 20 events. In the first simulation, where R = 5 h, the simulated hydrograph was representative of direct runoff according to the physical interpretation of the storage time; here, the fastest response of the system occurred. The peak discharge was two orders of magnitude greater than the observed values, and the recession limb exhibited a high slope according to the behavior of the considered runoff process. Among the available simulations using R values falling within the range of a fast return flow (R = 13, R = 15, and R = 20), the simulation with

R = 20 h appeared to provide better prediction modeling for both the peak discharge evaluation and the first part of the falling limb (fast and direct flow). The simulated delayed recession limb (R = 150 h), instead, was underestimated here with respect to the observed limbs because the applied R value was not high enough to also consider the deep component. According to [29,32], this R value indicates that the runoff component is generated from water released by means of soil pipe flows. The simulation with R = 50 and 150 h confirmed that these times indicated a fast and delayed return flow.

To test the model's reliability, mass balance hydrograph separation was performed following the procedure proposed in [29], using only the EC as tracers, along with field hydro-geomorphologic monitoring. The authors presented new mass balance models (MBMs) that, using a cascade approach, are able to separate a hydrograph into more than two runoff components and related mechanisms. Figure 5 compares the resulting rainfall–runoff model (RRM) derived by the HEC-HMS application and the runoff components derived by the mass balance model (MBM) using the end-members defined in [29] (Table 2).

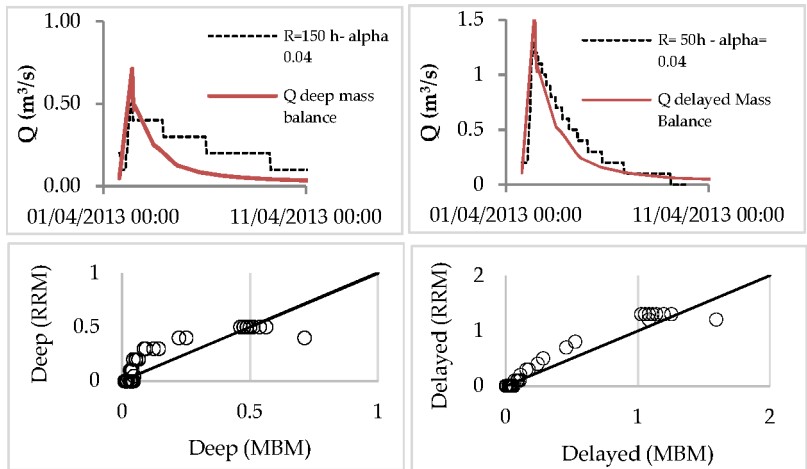

**Figure 5.** Comparison between the fast and the delayed return flow derived from the mass balance model (MBM) and the rainfall–runoff model (RRM).

A common and simple approach to evaluate models is to regress the predicted vs. observed values (or vice versa) and compare the slope and intercept parameters against the 1:1 line. In this case, in Figure 5, we plotted the fast and delayed flow estimated with the rainfall runoff model (RRM) versus the value derived with the MBM. For both, the peak discharge calculated with the MBM was clearly larger than that from the RRM procedure. Nevertheless, the fast return flow was well predicted, and the values on the rising and recession limbs were comparable between the two models.

The delayed component comparison exhibited a discrepancy between the two estimated components, but, considering the observed recession limb of the delayed component (Figure 4), the RRM gave a better prediction. In addition, comparing the volume production from the three components indicated that the delayed component (from the subsurface soils) provided about 30% of the total runoff, the fast return flow provided about 56%, and the direct component (as a Hortonian mechanism) provided 14% of the total runoff. These results were confirmed by previous studies on the same catchment [35], which highlighted that at the end of the rainy period, the system was saturated, and the hydrological response was sustained by groundwater ridging along the riparian corridors as the main components.

The results of this research can be compared to the results in [4], as the event simulated herein had an API30 (108 mm) comparable to that in [4]. Moreover, the studied catchments are headwaters and are characterized by a Mediterranean climate. Where a positive linear relation was found between the API and the initial water depth in the aquifers in [4], in

this study, the high API30 predicted deep and shallow aquifer saturations. For this reason, the ridge and nose contributions were not considered for modeling the components, and the predicted hydrograph was found to be adequate for this assumption. Indeed, the study in [4] demonstrated that the ZOB with a lower soil depth has a faster flow component than that with a higher soil depth. Our results confirmed that the fast component was the main component of Ciciriello for the simulated event in which the catchment is saturated. Indeed, under very wet conditions, most ZOBs become hydrologically linked to the channel system, so preferential flow systems expand and significantly enhance the subsurface flow, and overland flow contributions from the riparian zone become trivial [3].

The catchments, therefore, are characterized by a large storage capacity due to the delayed component, which may be useful for water resource planning and management in addition to deep storage.

The results of the lumped simulation confirmed the hydrologic interpretation of the $\alpha$ mean values in the previous section. The best simulation was obtained using $\alpha = 20$ h, which accurately predicted the peak discharge and direct runoff, but poor performance was obtained for the simulated slow components.

Good results were also obtained from the simulation of the fast and delayed return flow, which may be suitable for simulating the delayed ($\alpha_4 = 150$ h) and fast components ($\alpha_3 = 50$ h) in the distributed model. These results confirmed the hydrologic significance of the calculated $\alpha$ value, which can be used to predict the hydrologic parameters for distributed modeling.

### 3.3. Distributed Rainfall–Runoff Simulations

The distributed models provide a robust hydrologic simulation tool by dividing the catchment into sub-basins. In particular, the HEC-HMS models provide software routines to divide the catchment into sub-basins using ArcHydro as an ArcMap application. The subdivision of the catchment was performed starting from the Regional Technical Map of Campania (CTR) at a scale of 1:5000 by converting the map through a triangular irregular network (TIN) in a Digital Elevation Model with a 5 $m^2$ cell size, which was used for the flow accumulation derivation. The contributing area, as an input parameter required for the sub-basin delineation, was based on the numbers of cells in which the drainage begins and was set to 266 cells, a value obtained from the calibration, to match the derived drainage network to the river depicted in the CTR.

On the basis of the map in Figure 6, the geometric parameters for the sub-catchment and the relative river network were made available and suitable for the tc calculations by assuming Equation (3). In Table 6, the results for the time of concentration calculated with Equation (3) are reported for each sub-basin.

The time of concentration here ranged between 0.06 h, detected in the sub-basin with the minimum $\Delta z$, and 0.38 h. All the sub-basins had a tc greater than the whole catchment according to the length of the catchment's main channel, which was one order larger than the sub-basin's $L_a$.

Figure 6 illustrates the sub-basins obtained in the ArcGis environment. This figure shows the suitable map provided in the HEC-HMS model. Each symbol here has a specific hydraulic significance for the parameter assignment.

About 50% of the Ciciriello catchment featured fast return flow components, with 25% featuring delayed return flow components; only minor contributions to the runoff were derived from direct and deep percolation (13% and 12%, respectively).

To predict the discharge hydrograph at the outlet of the sub-basins, the R and $\alpha$ parameters must be estimated. These parameters are unknown at the sub-basins because they are ungauged, and there is a lack of hydrologic data suitable for modeling.

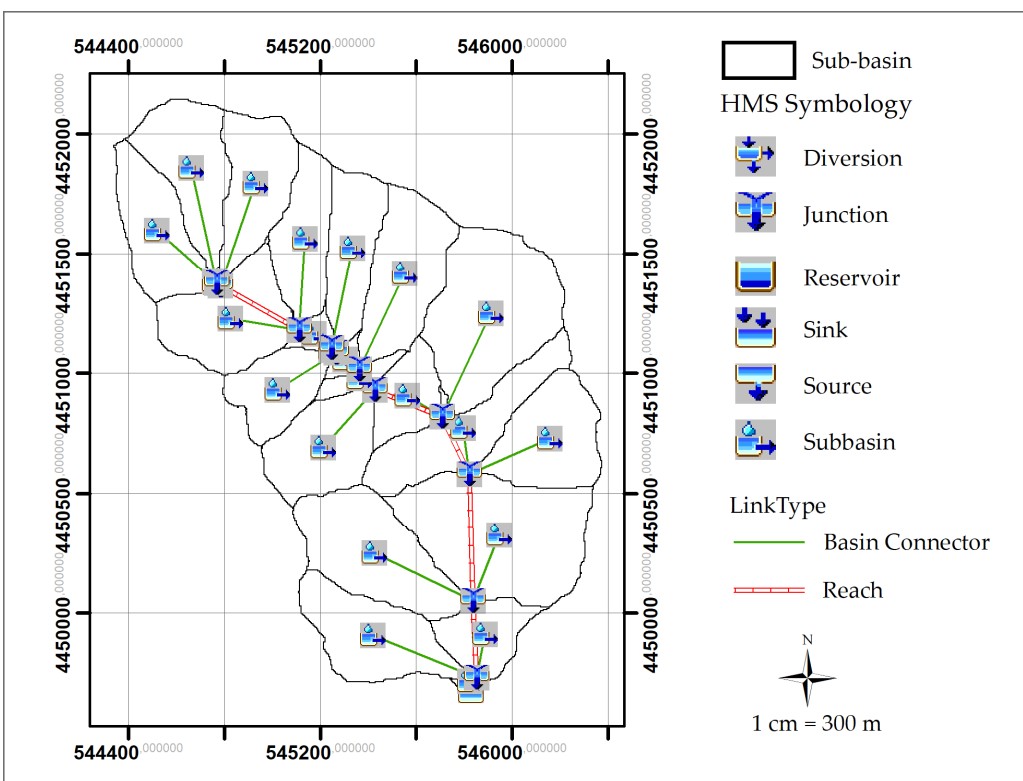

**Figure 6.** Ciciriello catchment divided into sub-basins for the distributed simulation.

### 3.3.1. Hydro-Geomorphologic Map

The hydro-geomorphologic map was developed following the procedure in [26]. The hydro-geomorphologic map (HGmM) (Figure 7) was derived in the GIS environment using the ArcMap software starting from the object-based geomorphologic units in Figure 2 and merging this map with the hydrogeological complexes featuring medium permeability.

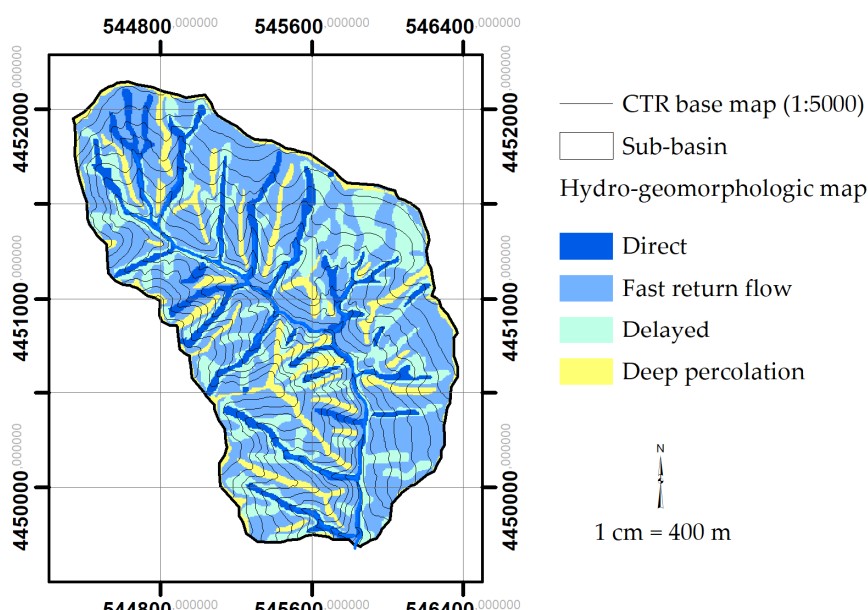

**Figure 7.** Object-based hydro-geomorphologic map of the Ciciriello catchment.

Therefore, Equation (4), which relates the physiographic catchment characteristics such as the geology, soils and geomorphology with the hydrological parameters to facilitate their calculations, was applied using the HGmT map. To this end, the hydro-geomorphologic indices (HGmI) were calculated at the outlet of each sub-basin.

Table 7 summarizes the HGmI for each runoff process individuated in the HGmT map.

**Table 7.** HGmI evaluated at the outlet of Ciciriello's sub-basins for each component (deep percolation, delayed and fast return flow, and direct runoff).

| Sub-Basin | $HGmI_{deep}$ | $HGmI_{delayed}$ | $HGmI_{ast}$ | $HGmI_{direct}$ |
|---|---|---|---|---|
| W240 | 0.11 | 0.04 | 0.60 | 0.25 |
| W250 | 0.15 | 0.20 | 0.48 | 0.17 |
| W260 | 0.11 | 0.32 | 0.46 | 0.12 |
| W270 | | 0.37 | 0.35 | 0.28 |
| W280 | 0.11 | 0.24 | 0.55 | 0.10 |
| W290 | 0.09 | 0.20 | 0.60 | 0.11 |
| W300 | 0.10 | 0.41 | 0.30 | 0.19 |
| W310 | 0.11 | 0.44 | 0.29 | 0.15 |
| W320 | 0.17 | 0.22 | 0.52 | 0.10 |
| W330 | 0.06 | 0.41 | 0.43 | 0.11 |
| W340 | | | 0.42 | 0.58 |
| W350 | 0.19 | 0.16 | 0.37 | 0.28 |
| W360 | 0.05 | 0.02 | 0.78 | 0.15 |
| W370 | 0.15 | 0.29 | 0.51 | 0.05 |
| W380 | 0.10 | 0.25 | 0.51 | 0.14 |
| W390 | 0.19 | 0.21 | 0.52 | 0.08 |
| W400 | 0.25 | 0.31 | 0.29 | 0.16 |
| W410 | 0.09 | 0.21 | 0.62 | 0.07 |
| W420 | 0.12 | 0.17 | 0.63 | 0.08 |
| W430 | 0.10 | 0.25 | 0.53 | 0.13 |
| W440 | 0.06 | 0.18 | 0.69 | 0.06 |
| W450 | 0.16 | 0.28 | 0.43 | 0.13 |
| W460 | 0.01 | 0.37 | 0.41 | 0.21 |

The HGmT map shows that the large contribution to the runoff generation in the sub-basins was derived from the fast return flow, which is justified by the large presence of soil pipes [30] in forested soils. The sub-basins with the lowest values of direct HGmI were W370 and W390, which act as interfluves and triangular facets in the form of planar surfaces with broad bases and upward-pointing apexes [47]. For the case study, the triangular facets were not characterized by the presence of a river network. Therefore, the main contribution to the runoff generation was related to the excess saturation runoff process. The sub-basins W410, W420, and W440 also had a low direct HGmI. These three basins are localized on the left side of the catchment, where mass movement was detected during geomorphologic surveys, thereby obtaining a large and fast HGmI that ranged between 50 and 70%. The largest contribution from the deep percolation came from the headwaters (0.11) and the small sub-basins (0.25). The calculated indices were very sensitive to the areal extension of the basin. For this reason, small catchments at heights as low as that of W400 exhibited higher values for deep HGmI than the headwater basins (W240, 250, and 260). The largest value of the delayed HGmI was 0.44 at the W310 basin, followed by 0.41 at the W300 and W330 sub-basins, whose contributions were derived mainly from the colluvial hollow and the zero-order basins.

### 3.3.2. Hydro-Geomorphological Rainfall–Runoff Simulation

The hydro-geomorphologic simulation was performed by assuming the hydrologic behavior of the sub-catchment according to the HGmT map and using Equations (4) and (5) for the R and $\alpha$ calculations, respectively.

Three simulations were performed. In the first simulation, only the $\alpha$ value was calculated with Equation (5), and the R was assumed to be constant for all the sub-basins and equal to 20 h. The $t_c$ values were calculated with the Giandotti formula and are listed in Table 4. The second simulation used the same $\alpha$ and $t_c$ values from the first simulation as the input. Only the R was calculated following Equation (4) as the weighted mean with respect to the HGmT.

The third simulation used the R calculated from the main runoff component (the largest HGmI in the sub-basin), and the other parameters, $\alpha$ and tc, were the same as those in the first simulation. For the three simulations, Figure 8 compares the observed versus the simulated discharge hydrographs.

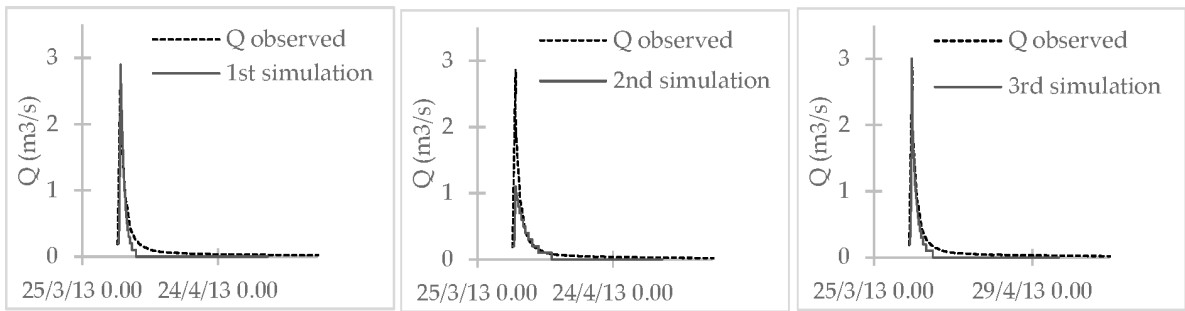

**Figure 8.** Comparison between the observed and simulated hydrographs for the first, second, and third simulations.

Figure 8 shows that the first and third simulations performed best. The second simulation, derived from the weighted mean of R (Equation (4)), underestimated the peak discharge; otherwise, the best simulations of the delayed and the deep percolation were obtained. For optimizing the interpretation of the resulting hydrographs, the common predicted regression vs. the observed values was used, and a comparison between the slope and intercept parameters against a 1:1 line was performed, as shown in Figure 9.

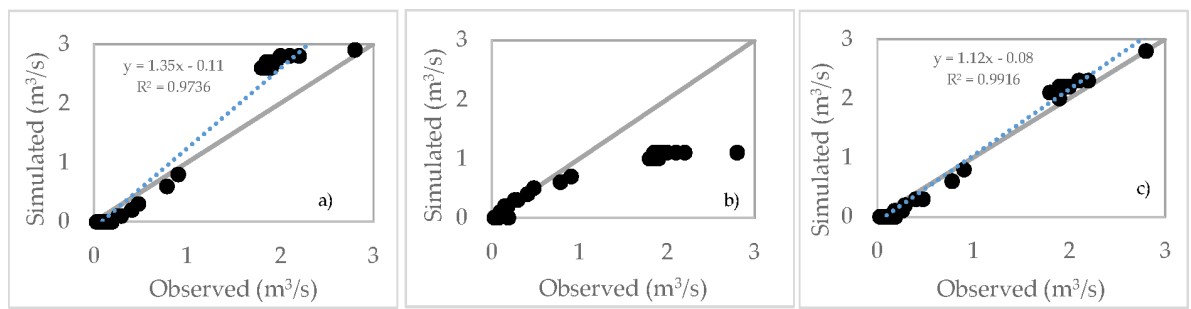

**Figure 9.** Plot of predicted against observed discharge hydrograph using the first (**a**), the second (**b**), and the third simulations (**c**). Grey line: 1:1 line; dotted blue line: linear relation between the simulated and observed hydrographs.

In Figure 9, a good relationship can be seen between the observations and models for the first and third simulations. The slope of the straight line of the third simulation was close to 1, and the origin was near zero more so than in the first simulation. The second simulation confirmed a linear relation for the low flow, but large prediction errors were obtained for the high flow. This result could be derived from an R value that, when calculated using Equation (4), was comparable with the mean storage time estimated at the main station for the delayed return flow. Indeed, the applied input R values ranged from a minimum of 23 to a maximum of 83 h, all included in the range of the fast and delayed return flow calculated from the observed hydrographs. Conversely, the R used in the third simulation used 0.8 h as a minimum value and 66 h as the maximum. Respect to the second simulation, in the third simulation was employed the lowest R values enhancing the direct runoff, thus optimizing the peak discharge and the direct source. An unexpected result

was derived from the first simulation, where only the recession constant $\alpha$ was calibrated with the HGmI, but good performance was obtained for the model. The input R value used was 20 h, which was comparable to the mean R value assumed for the third simulation (24 h) that gave the best performance in the fast responses of the predicted hydrograph.

To objectively evaluate the performance of the predictions, Table 8 summarizes the statistical indices derived from Equations (6)–(10).

**Table 8.** Indices for the performance of the models. (MAE: Mean Absolute Error; MSE: Mean Squared Error; RMSE: Root Mean Squared Error; NSE: Nash–Sutcliffe Efficiency coefficient; and d: Index of agreement.

| Simulation | MAE | MSE | RMSE | NSE | d |
|:---:|:---:|:---:|:---:|:---:|:---:|
| 1 | 0.22 | 0.35 | 0.59 | 0.64 | 0.86 |
| 2 | 0.25 | 2.76 | 1.66 | 0.32 | 0.78 |
| 3 | 0.10 | 0.01 | 0.10 | 0.95 | 0.93 |

The goodness-of-fit indices confirmed that the third simulation was the best prediction method. The MSE was optimized from 0.35 to 0.01 for the third simulation, while the index of agreement d moved from 0.86 to 0.93. Improvements were even more evident, with NSE showing an increase of about 1.5 times from the first to the third simulation, as well as with the RMSE, which showed a decrease of about 6 times.

It follows that the simulated parameters for the lumped models were successfully transferred to the 23 un-gauged sub-basins using the HGmT map, which adequately represented the hydrological response of the Ciciriello catchment.

The results obtained for each sub-basin highlight that the main contributions to streamflow were derived from the W420 (11%) and W330 (9.6%) sub-basins, followed by the W430 (7.4%), W400 (7%), W450 (5.8%), and W310 (6.2%) sub-basins. The lowest contributions were derived from the W270, W320, W340, W360, and W460 sub-basins, which augmented the stormflow with a percentage less than 1% due to their small areal extensions. Despite their lower extensions, W270 and W340 augmented the stormflow with direct runoff (alpha=17). Consequently, according to Kim et al. [5], the discharge increased and declined rapidly. Sub-basins W420 and W330, which provided the largest increase to the stormflow, mainly contributed a fast return flow originating from the hillslopes according to the lumped model results (56%). Other sub-basins, i.e., W300, W310, and W420, helped increase the streamflow with the fast component, despite the main HGmT being the delayed unit. For these sub-basins, the $\alpha$, derived from the simulated hydrograph, was within the range of the fast return flow (0.8–0.36 1/d). These findings are supported by other studies on Mediterranean headwater catchments in wet conditions; during several storms, in addition to larger storms, delayed and fast components occurred [5]. A study by Latron et al. (2008) [48] on two small Mediterranean catchments also verified that under wet conditions, the main contributions derived from the excess saturation processes favored a prone and more extended saturation of the downslope areas, which, in turn, increased the hydrological response. In addition, as wetness increased, the "threshold response" concept of Sidle was verified [3]. After several storms, the water filled the shallow soil and the sub-surface soil matrix, and the runoff began to self-organize and expand with preferential flow pathways that facilitated fast drainage [3]. According to Kim et al. [5], for the wet event simulated here, the shallow groundwater flow contributed to the baseflow, and only surficial and sub-surficial tanks contributed to the runoff generation.

## 4. Conclusions

This work focused on the hydro-geomorphologic water balance of a Mediterranean headwater catchment at event scale and it included three main steps: calibrating the hydrologic parameters, physical-based rainfall–runoff modeling, and testing the adopted procedures.

Very often, the lack of information on the forested headwater catchment makes it difficult to properly calibrate the parameters and study their hydrologic behaviors (e.g., due

to a reduced areal extension in the catchment, leading to little interest among institutions, as well as the difficulty in reaching such areas to apply an adequate monitoring system). The use of easily obtainable physical parameters plays a central role in rainfall–runoff modeling, especially distributed parameters. Therefore, in the scientific literature, formulations exist for hydrological parameter evaluations through the use of the physiographic characteristics of the catchments, especially for ungauged catchments. In this study, the hydrological pathways, source areas, and flow generation processes within the various hydrogeomorphic components (e.g., noses, hollows, riparian areas, and hillslopes) were used for calibrating the hydrologic parameters and determining the hydrologic behavior of the catchment under wet conditions. These components play a central role in the soil–water balance, connecting the soil type and thickness to the hydro-geomorphotype, characterizing the flow pathways, and quantifying the hydrograph storm runoff component.

The object-oriented hydro-geomorphologic method proposed in this paper started from the hydro-geomorphic paradigm of Sidle [3] and used the object-based hierarchical and multiscale geomorphologic mapping of Dramis et al. [27], which is able to translate the geomorphological objects from larger to smaller scales, and vice versa. This object-oriented hydro-geomorphologic model is a new approach for the assessment of hydrological model behavior, especially in upstream ungauged sub-basins within catchments with unique gauged outlets.

The resulting goodness-of-fit indices revealed advanced knowledge of the geomorphologic processes occurring in the catchment. The interpretation of these processes from a hydrologic point of view can yield good calibration of the parameters for model application. The availability and quality of hydro-geomorphologic knowledge and maps can provide effective data calibration, especially in un-gauged basins, as well as contributing to water resource management in the broader context of water scarcity.

The present method is easy to apply and inexpensive because it is based on geological and geomorphological knowledge of the catchment that can be derived from published high-resolution geologic maps or from traditional expert-based geomorphological maps. Modern geomorphologic studies alongside the object-based geomorphologic mapping used herein will help make this method more objective and repeatable in other similar catchments.

In particular, the model applied in this study could be used to assess the hydrologic responses to other similar Mediterranean catchments or those under diverse climate conditions after adequate hydro-geomorphologic map production, hydro-chemical monitoring, and object-based hydrograph separation.

**Author Contributions:** Conceptualization, A.C. and D.G.; methodology, A.C.; software, A.C.; validation, D.G.; formal analysis, A.C. and D.G.; investigation, D.G.; data curation, A.C. and D.G.; writing—original draft preparation, A.C.; supervision, D.G.; funding acquisition, D.G. All authors have read and agreed to the published version of the manuscript.

**Funding:** This research was funded by the University Research Annual Founding (FARB) GUIDA 2017 and 2018 and by the financial contributions of the C. U. G. RI. (InterUniversitary Consortium, denoted as the Research Center for the prevention of Great Risks).

**Institutional Review Board Statement:** Not applicable.

**Informed Consent Statement:** Not applicable.

**Acknowledgments:** The authors are very grateful to Lovisi Pasqualino for field measurements. Thanks also to Eng. Mauro Biafore for the rainfall data from the Campania region Monitoring System. A special dedication to **Fabio Rossi**, Hydraulic engineering at the University of Salerno. He was a founder and the first Director of C.U.G.RI. and the ideator of the VAPI method.

**Conflicts of Interest:** The authors declare no conflict of interest.

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
