# Peer review of "Hydro-Geomorphologic-Based Water Budget at Event Time-Scale in A Mediterranean Headwater Catchment (Southern Italy)"

_hydrology, doi:10.3390/hydrology8010020_

Round 1

Reviewer 1 Report

Taking the Ciciriello catchment, located in the southern part of Campania Region (Southern Italy), as key study area, using the HEC-HMS model and 16 object-based hydro-geomorphological class features, this paper puts forward one procedure to calibrate the 15 hydrologic parameters for a water budget at event scale. This research includes three steps, calibrating the hydrologic parameters, physical based rainfall-runoff modelling, testing adopted procedures.

Some questions:

  • The citation format of the references in this article is correct or wrong? for example, line 61, [9] introduced …., why [9], not specific author?
  • The questions and objective of this article need to reinforce.
  • Line 87, the average annual rainfall (raifall?) is around 1.400 mm, 1,400 mm or 1.400mm?
  • line 110, at 1:5.000 scale?
  • line 113, scale 0,275 or 0.275?
  • all paper, check use of decimal points and ten percentile, for example, in Table 4, how many is of t?
  • in table 6, the numbers of this tables are very confused, please check.
  • The conclusion needs to rewrite.
  • Extensive editing of English language and style are required for whole article.

Reviewer 2 Report

Review of paper by Cuomo & Guida submitted to Hydrology:

General Comments:

This is a very interesting paper on an important catchment hydrology topic that has not been investigated by many researchers in such a manner. The methods and science involved in this work are quite solid and the modelling approach has unique aspects. I have outlined some omissions below, but I think these can be easily incorporated by the authors. They need to look at the Sidle et al., (2011) paper on hydrogeomorphic modeling in a small forest catchment that is similar to their study, but that takes a different approach to modelling using a multi-tank model and a kinematic wave model. Thus, from a scientific perspective, the paper is quite good and original once the issues that follow are addressed by the authors.

The major issue with the paper is the English presentation. I started to edit the Abstract, but I did not have time to edit the entire paper as there are many problems and this would be a very big task. In a few places, however, I made some suggestions. For one thing, the authors need to avoid using one sentence paragraphs that appear in many places. They also need to take care of referencing – generally, it is not appropriate to begin a sentence with a ‘numbered reference’. e.g., [4] also described a catchment… Please see the guidelines for this journal. While the paper was readable, it needs a very thorough review by a native English writer to help correct the many issues within the paper. Once this is done and the comments below are addressed, I think this will make an excellent contribution to the hydrological literature.

Introduction:     The Introduction is lacking some key up-to-date references. Aside from their own references, all papers cited in the Introduction have been published prior to 2000. Many new studies in hydrogeomorphic response have been published in the past two decades and some of these should be discussed herein. A few possibilities include:

Borga et al., 2014 in Journal of Hydrology

Clarke et al., 2008 in J. American Water Resour. Assoc.

Francipane et al., 2012 in Catena

Molla et al., 2016 in Journal of Hydrology

Sidle et al., 2000 in Hydrol. Process.

Sidle et al., 2011 in Water Resources Res.

Kim et al., 2011 in Hydrological Research Letters

Materials & Methods section:    You introduce the storage term (R) in lines 212 and 226-231. Can you please add a sentence or two describing the physical relevance of this R term for the various storage components?

Results section (3.3.2 & 3.3.3):   These are very interesting findings. What I recommend, however, is that the authors take a closer examination of the components of various catchments that contributed to different types of flow phenomena. For example, what hydrological stormflow mechanisms (i.e., quick return flow, direct runoff, delayed return flow and deep percolation) dominated (and which ones were not important) in various parts of the catchments (i.e., hillslopes, hollows, noses, ridges, and riparian corridors). I think you can do this with the results you have shown, and this would add much value to this paper. If you look at the two related papers by Sidle et al. (2000 in Hydrological Processes which you already cited, and 2011 in Water Resources Research (and also the Kim et al., 2011 paper in Hydrological Research Letters), where they modeled these pathways using a multi-tank model), you can compare your findings of dominant flow paths in various hydrogeomorphic features (hollows, riparian corridors, hillslopes, etc.) with the finding from these other studies that are also conducted in a temperate forested catchment. I think this comparison would substantially improve your modelling assessment and the associated interpretations.

Conclusion:        In the first short paragraph (lines 510-513), I would argue that in addition to the parameters at the outlet of each sub-catchment, it is necessary to have data on the hydrological pathways and flow generation processes within the various hydrogeomorphic components (e.g., hollows, riparian areas, hillslopes).  I think in your study, you actually did this, but it should be explained better here; you start to do this in the second paragraph. Please try to merge these concepts in a better way. In line 518 you state this method is new; your recession limb approach is indeed new as far as I can see, but the Sidle et al (2011) paper in Water Resources Research was the first (I believe) to specifically target various hydrological pathways in different parts of a small catchment based on the earlier field studies and conceptualizations (Sidle et al., 2000).

This paper was of great interest to me and I hope to see it published soon. Also see my edits in the attached text.

Best regards,

Reviewer 3 Report

Dear authors, I have carefully reviewed your paper and found it very interesting. I think that it is a good advance to model some hydrological characteristics of a Mediterranean catchment. However, you should improve some issues before accepting this paper. Attached, you can find some comments in the pdf. I consider that there are a lot of parts of the ms without references. I won´t recommend you anyone, but please, the intro and discussion are almost written without references, and this is not correct. The climate and geological survey need references. The figures must be homogeneous (same scale, grids, font types, avoid the use of italics, etc.). The discussion is only using papers about the same catchment and is not specific. Please, add more study cases and areas.
